# Co-Application of Milk Tea Waste and NPK Fertilizers to Improve Sandy Soil Biochemical Properties and Wheat Growth

**DOI:** 10.3390/molecules24030423

**Published:** 2019-01-24

**Authors:** Tanveer Ali Sial, Jiao Liu, Ying Zhao, Muhammad Numan Khan, Zhilong Lan, Jianguo Zhang, Farhana Kumbhar, Kashif Akhtar, Inayatullah Rajpar

**Affiliations:** 1College of Natural Resources and Environment, Northwest A&F University, Yangling 712100, China; alisial@nwafu.edu.cn (T.A.S.); luxy0803@163.com (J.L.); mnkses777@gmail.com (M.N.K.); zll199101@gmail.com (Z.L.); 2Department of Soil Science, Sindh Agriculture University, Tandojam 70060, Pakistan; irajpar@yahoo.com; 3College of Resources and Environmental Engineering, Ludong University, Yantai 264025, China; 4College of Agronomy, Northwest A&F University, Yangling 712100, China; kumbarfarhana@yahoo.com; 5Department of Agronomy, The University of Agriculture, Peshawar 25000, Pakistan; Kashtoru@gmail.com

**Keywords:** milk tea waste, wheat growth, nutrients uptake, root traits, soil enzyme assays

## Abstract

Desert soil is one of the most severe conditions which negatively affect the environment and crop growth production in arid land. The application of organic amendments with inorganic fertilizers is an economically viable and environmentally comprehensive method to develop sustainable agriculture. The aim of this study was to assess whether milk tea waste (TW) amendment combined with chemical fertilizer (F) application can be used to improve the biochemical properties of sandy soil and wheat growth. The treatments included control without amendment (T1), chemical fertilizers (T2), TW 2.5% + F (T3), TW 5% + F (T4) and TW 10% + F (T5). The results showed that the highest chlorophyll (a and b) and carotenoids, shoot and root dry biomass, and leaf area index (LAI) were significantly (*p* < 0.05) improved with all amendment treatments. However, the highest root total length, root surface area, root volume and diameter were recorded for T4 among all treatments. The greater uptake of N, P, and K contents for T4 increased for the shoot by 68.9, 58.3, and 57.1%, and for the root by 65.7, 34.3, and 47.4% compared to the control, respectively. Compared with the control, T5 treatment decreased the soil pH significantly (*p* < 0.05) and increased soil enzyme activities such as urease (95.2%), β-glucosidase (81.6%) and dehydrogenase (97.2%), followed by T4, T3, and T2. Our findings suggested that the integrated use of milk tea waste and chemical fertilizers is a suitable amendment method for improving the growth and soil fertility status of sandy soils.

## 1. Introduction

The continuous increase of the global population puts great pressure on the environment and food production. To meet crop production, food for future generations will put a strain on soil and water resource security in the coming decades [1]. China has the world’s largest population, with approximately 25% of the world’s population [2]. To maintain the food demand of a continually increasing population trend in China, as well as throughout the world, the usage of a high rate of fertilizers for high crop production and additional soil resources need to be accepted and employed. A previous study [3] established that the soils of arid regions are considered as having low water, carbon, and nutrient contents [4]. Soil sandification is a serious issue due to the increasing rate of around 50,000 to 70,000 km^2^ of area every year, and the economic losses expected amount to approximately 42 billion USD [5]. Taklimakan is the biggest desert in China and second largest shifting desert in the world, which is located in northwestern China, covering an area of 338,000 km^2^ [6]. The Chinese government also faces economic losses around 54.1 billion RMB every year due to sandy desertification [7]. The sandy soils have low soil nutrient contents, a negligible content of carbon (C) stock and alkaline in reaction [1]. Plant nutrients are major components of crop growth and, out of 16 essential elements, nitrogen (N), phosphorus (P), and potassium (K) are essential elements for crop growth [4,8]. After C, N is the element most required in large quantities by crops, because it is an integral constituent of protein, chlorophyll and other metabolic reactions [9]. Therefore, high doses of N fertilizers are required, because of the increasing grain demand of the increasing trend in population [10], causing economic and environmental losses; specifically, in sandy soil, there are high N losses due to leaching and low nitrogen use efficiency [1,9]. To document these, different approaches had been conducted using different levels of N fertilizers and organic amendments [8,10,11]. However, this could be used as an additional source for cultivation and could cover the food demand for future generations, and the soil quality could be improved with organic amendments, and the N for crop demand could be supplied under different soil textures [1,4,12]. The high rate of chemical fertilizers, especially N, improved the light carbon fraction of soil C breakdown by microorganism [13]. Phosphorus and potassium are also important for plant growth and physiological reactions for low fertility soil [14,15]. However, the overuse of chemical fertilizers obviously declines the soil quality and crop yield [14,16]. The co-application of organic and inorganic amendments is a sensible fertilization strategy for sustainable agriculture and the environment, especially in low fertility soils [10]. The application of chemical fertilizer alone is not effective, but application with organic manure might be a better option for maintaining soil organic carbon stock, crop growth, and root growth traits [17].

The root apparatus is a bridge between the aboveground growth of plants and the soil matrix; a plant uptakes nutrients and water from the soil by its root network system. The root faces a large number of soil environmental problems such as water stress [4], soil hardness, low fertility [8], metal toxicity, salinity, and waterlogged conditions, which are big problems for root growth [18]. Root morphological traits including root length, surface area, and radius express a plant’s capacity to compete for soil nutrients [11]. Soil NH_4_^+^-N could be submissively taken up by roots [19], which can directly uptake the wide range of NH_4_^+^-N concentrations for plant growth [20]. Soil nitrate (NO_3_^−^) is an easily available mobile form in the xylem and can also be stored in the vacuoles of the root, shoot, and storage organs. Root length is generally recognized to be proportional to water and nutrient attainment, while biomass accumulation is correlated to root diameter [21]. Photo-synthetically secure carbons to soil organic matter (SOC) fractions are also transported [2,22]. The root length ratio (RLR) plays an important role under soil stress environments [23] because RLR escapes algometric effects [24]. Root density and root fineness are considered major components of root growth [23]. Organic amendments with chemical fertilizer application improve the growth, nutrient uptake and soil quality [4,22,25].

Is milk tea waste a better option for the root growth traits, plant growth, soil quality and nutrient contents of sandy soils? Tea consumption throughout the world, such as in Pakistan, Bangladesh, and India, occurs with a small amount of milk, sugar, and water, while China, Japan, United Kingdom, Ireland, and Canada use a considerable quantity of milk [26]. Should milk tea waste be used for soil modification concerning biological properties such as enzymatic activities? Soil enzymatic activities, as major biocatalysts involved in all biochemical reactions including the microbial life cycle and metabolism, break down and decompose soil organic matter and residues, and degrade perilous organic contaminants and encourage nutrient cycling [27]. The application of organic amendments improved the soil fertility, wheat crop growth, soil microbial activity and the physical properties of sandy soils [4]. To our knowledge, there is no evaluation of milk tea waste being used as a soil amendment for plant growth, root growth traits, and soil quality. In the present study, we used kitchen waste (milk tea waste) applied as a nutritional source for plants and sandy soil combined with chemical fertilizers. Milk tea waste contains nutrients such as N, P, K, and a low C:N ratio as compared to other organic amendments. It could be a better amendment for sandy and low fertile soils.

The objectives of our study were (1) to evaluate the application of milk tea waste on root and plant growth and the uptake of N, P, and K contents in wheat; and (2) to quantify the effects of milk tea waste on the biochemical properties of sandy soil.

## 2. Results

### 2.1. Effect of Milk Tea Waste (TW) Application on Plant Growth

The incorporation of the amendment combined with chemical fertilizers within sandy soil significantly (*p* < 0.05) increased chlorophyll content, plant height (cm), leaf area (cm^2^), shoot and root dry biomass (g·pot^−1^) over the control (Figure 1 and Table 1). The chlorophyll (a and b) and carotenoid contents were also significantly (*p* < 0.05) enhanced with soil amendments of TW application rates (Figure 1A–C). The application of T4 treatment significantly improved the chlorophyll (a and b) and carotenoid of wheat among all treatments (49.6, 30.9, and 39.3%) over the control. The highest plant height was recorded for T4 (30 cm) and lowest for the control (18 cm), while chemical fertilizer alone and combined with TW 10% did not significantly affect plant height as compared to the control treatment. The highest shoot and root dry biomass were obtained for T4 treatment. The percentages of the increase in shoot and root dry biomass for T4 were 71.5% and 53.1%, for T3 were 56.9% and 45.2%, for T5 were 42.8% and 25.8% and for T2 were 39.8% and 16.7%, respectively, compared with the control.

### 2.2. N, P, and K Contents in Shoot and Root

The total nitrogen (N), phosphorus (P) and potassium (K) uptake by wheat plants were significantly (*p* < 0.05) improved after amendment application (Figure 2A–C). N uptake in the shoot and root was significantly (*p* < 0.05) increased by TW incorporation combined with chemical fertilizer (Figure 2A). A higher N uptake for the shoot of 68.9% and root of 65.7% were recorded in T4, respectively, compared to the control. A similar trend in P uptake in the shoot and root (8.2 and 5.8 g·kg^−1^) were obtained for T4, and the lowest values of 2.9 and 1.9 g·kg^−1^ were observed for the control. The P uptake improved with the addition of TW + F as compared to the control. However, the root P uptake trend was different the maximum percentage increase for T4 (67.4%), followed by T3 (52.6%), T2 (40.0%), and T5 (14.6%), respectively, higher than the control. The shoot and root K uptake was also significantly (*p* < 0.05) enhanced by the incorporation of TW and the sole application of chemical fertilizer (Figure 2C). Chemical fertilizer application alone improved the K uptake in the shoot by 22.6% and in the root by 28.8% over the control, while a, higher K uptake was noted for T4 by 57.1% in the shoot and by 47.4% in root in comparison to the control.

### 2.3. Root Traits

Root traits were significantly (*p* < 0.05) influenced by the different milk tea waste (TW) rates combined with inorganic fertilizers under sandy soil environment (Figure 3 and Table 2). The total root length, surface area, root volume, and root diameter (841 cm, 102 cm^2^, 1.2 cm^3^, and 0.53 mm) were significantly increased for T4 treatment, and were lowest (201 cm, 27.2 cm^2^, 0.36 cm^3^, and 0.35 mm) in the control (Figure 3).

The root density (RD), root length ratio (RLR), and root fineness (RF) (Table 2) were significantly (*p* < 0.05) improved by TW amendment combined with chemical fertilizers, but the root mass ratio (RMR) was not improved by amendments (Table 2). Chemical fertilizer alone did not significantly increase the values of those root parameters. The RD, 0.43 to 1.31 cm·g^−1^, RLR, 674 to 1145cm·g^−1^, and RF, 457 to 956 cm·cm^−3^ were improved among all treatments, while T4 significantly improved the root traits. However, the RMR trend was opposite, with the maximum observed for the control (0.52 g·g^−1^) and the lowest values for the T4 (0.40 g·g^−1^) treatment.

### 2.4. Soil Chemical Properties

As shown in Figure 4, the co-application of milk tea waste (TW) and chemical fertilizers for soil amendment had significant (*p* < 0.05) effects on soil pH, soil organic carbon (SOC), NH_4_^+^-N, NO_3_^−^-N, available phosphorus (AP) and available potassium (AK). Except for the pH, the soil parameters of all TW treatments including chemical fertilizer treatment increased compared with the control after eight weeks of wheat harvested. Soil pH values for all treatments ranged from 7.6 to 6.1 (Figure 4A), and decreased with an increasing rate of TW, with the maximum reduction of pH over the control treatment in T5 (1.4 units) followed by T4 (1.3 units), T3 (1.2 units), and T2 (0.3 units), respectively. SOC was increased following the combined application of TW with inorganic fertilizers (F). The highest SOC content was recorded under T5 (25 g·kg^−1^) and lowest under control (3.4 g·kg^−1^) treatment (Figure 1B). SOC content increased the percentage in the following order: T5 (86.3%), T4 (79%), T3 (67%), and T2 (20%) as compared to the control. The content of SOC increased with an increasing rate of TW amendments. Soil mineral nitrogen (NH_4_^+^-N and NO_3_^−^-N) concentrations indicated significant variations among different treatments (Figure 4C,D). After the wheat was harvested, the soil NO_3_^−^-N value in all treatments was greater than the control (1.8 to 4.2 mg·kg^−1^). Soil NO_3_^−^-N content in milk tea waste treatments (corresponding to T3, T4, and T5) were observed to be significantly higher than the control only in T4, while T3 and T4 were not significantly different from the control. However, the soil NH_4_^+^-N trend was significantly increased with an increasing rate of TW amendments. The soil NH_4_^+^-N values among all treatments ranged from 1 to 19 mg·kg^−1^, which was lowest under the control and highest under T5.

The application of the amendments if TW combined with chemical fertilizer displayed a significant influence on soil AP and AK after the wheat was harvested (Figure 4E,F). The highest AP concentrations were noted for T5 (127 mg·kg^−1^), followed by T4 (65 mg·kg^−1^), T3 (34.4 mg·kg^−1^), T2 (21 mg·kg^−1^), and lowest for the control (12 mg·kg^−1^). The soil AK significantly increased following the increased application rates of TW combined with chemical fertilizer. The greatest AK concentrations were found in T5 (82.4%), followed by T4 (80.4%), T3 (52.2%), and T2 (31.4%), greater than the control.

### 2.5. Soil Enzyme Activities

The enzymes activities are attributed to the types of amendment and soil properties because of the interactive differences, and their adsorption and complexation with soil colloids. The soil urease, β-glucosidase, and dehydrogenase activities of different treatments were significantly (*p* < 0.05) influenced (Figure 5A–C). The urease activities significantly accelerated with increasing TW rates and also increased under chemical fertilizer treatment alone (Figure 5A). The maximum urease activities were noted for T5 (68 mg NH_3_-N g^−1^ soil 12 h^−1^), followed by T4 (40 mg NH_3_-N g^−1^ soil 12 h^−1^), T3 (17 mg NH_3_-N g^−1^ soil 12 h^−1^), T2 (12 mg NH_3_-N g^−1^ soil 12 h^−1^), and control (3.1 mg NH_3_-N g^−1^ soil 12 h^−1^). Likewise, β-glucosidase activities were significantly influenced after amendment application. The β-glucosidase activities between all treatments ranged from 169 to 875 mg *p*-nitrophenyl kg^−1^ soil h^−1^ (Figure 5B). Increased β-glucosidase activity in comparison to the control (T1) was measured for T2 (40.1%), T3 (72.2%), T4 (77.1%), and T5 (81.6%), respectively. The results indicated that soil amendments positively enhanced the dehydrogenase activities with increasing TW rates. The highest dehydrogenase activities were observed for the T5 (105 mg TPF kg^−1^ soil h^−1^) and the lowest for the control (3.0 mg TPF kg^−1^ soil h^−1^). The integration of TW and fertilizer treatments enhanced dehydrogenase activity over the control in the order of T5 (97.2%), T4 (96.4%), T3 (90.8 %), and T2 (46.0%).

### 2.6. Relationships among Root and Plant NPK, Soil Enzymes Activities and Soil Properties

Significant correlations between soil properties and soil enzyme activities are shown in Table 3. The Pearson correlation analysis revealed that soil properties (C, N, P, NO_3_^−^ and NH_4_^+^) had a positive correlation with soil enzymes. Three soil enzymes were significantly correlated when compared with soil properties except soil pH. However, soil pH was significantly negatively correlated with soil chemical properties and soil enzymes. There was no significant correlation between soil NO_3_^−^ and other soil chemical properties and soil enzymes (Table 3). The PCA showed clear differences in the root attributes and plant NPK content (Figure 6) and, in soil properties and enzyme activities (Figure 7) among the different treatments. All studied root parameters and plant NPK contents were clearly clustered into five well-differentiated groups. The first group contained the samples from T1 (control), the second group from T2 (chemical fertilizer), the third group from T3 (F + 2.5% TW), the fourth group from T4 (F + 5% TW) and the fifth group from T5 (F + 10% TW). The cumulative variance of contributions reached 92.6% (PC1 explained 84.7% and PC2 explained 7.9%) for root variables and plant NPK contents (Figure 6). Similarly, the cumulative variance of contributions reached 97.2% (PCA1 explained 94.9% and PCA2 explained 2.3%) for soil enzymes and soil properties (Figure 7). The analysis revealed that root traits, plant NPK, soil properties and soil enzymes were highly affected by chemical fertilizer and increasing milk tea waste (TW) treatments.

## 3. Discussion

### 3.1. Effects of Milk Tea Waste (TW) on Wheat Growth Traits

Our results showed that milk tea waste had significantly positive effects on wheat growth and root traits during the eight weeks period for the tested sandy soil. These improvements in wheat crop performance agree with recently published studies [4,8,28] and the established positive effects of organic amendments and chemical fertilizers on plant growth traits and soil properties, varying by oil textures and organic amendments. Our study documented that chlorophyll (a and b) and carotenoid contents, plant heights, leaf area indexes, as well as the shoot and root dry biomass were influenced by the organic amendment rate combined with chemical fertilizer. Our findings are consistent with [4,8,29], which attained a significant result of chlorophyll (a and b) and carotenoid content in a vegetable crop (Brassica) under the three different soils which were treated with bamboo biochar at a 5% rate. This may be associated with the slow release of nutrients from organic amendments at seedling to harvesting stages, which thus improved the soil fertility. Previous studies assessed that the co-application of organic and inorganic fertilizer amendments improved wheat growth, root traits, and soil biochemical properties [1,11,30,31]. In our study, chlorophyll (a and b) contents increased in the response of TW amendment, associated with N availability in the soil and making it easily available for the plants. Wheat plant heights, leaf area indexes, as well as the shoot and root dry biomass were significantly improved under TW + F amendment over the chemical fertilizer alone and control treatments (Figure 8). The increase in the chlorophyll contents of the wheat leaf displayed an increasing availability of nutrients due to the positive effect of TW amendment on growth parameters, while the high dose of TW (10%) displayed a reduction trend in all above parameters as compared to the 2.5% and 5% TW treatments. This might be ascribed to the large amount of TW as a source of organic matter, and facing the problem of a higher population of microbial activity and slowing the decomposition rate due to sandy soil environment. Our results agreed with previous study [4] that the high rate of organic of amendments had enhancing properties on life in the poor soils but initiation of functional and difficult in the biological process.

### 3.2. Impact of TW on Wheat Root Traits

We observed a significant increment in root growth under a sandy stress environment with the application of TW + F. After eight weeks, we observed that the highest root growth traits such as total root length, surface area, root volume, and diameter were improved for T4 over the control treatment. The chemical fertilizer amendment alone had no meaningful influence on root traits, while the combined application of TW and inorganic fertilizer significantly increased root growth. Our results are in line with [2], which evaluated that the sole application of chemical fertilizer was not effective for root growth traits, and [32] observed that organic amendments had significant impacts on maximum root development. The root length density is normally less than 1 cm·cm^−3^, included in the critical threshold level [33]. These findings support our results as the root length density values were greater than 1 cm cm^−3^ in T3 (1.2 cm·cm^−3^) and T4 (1.3 cm·cm^−3^) treatments. In the present study, TW + F amendments significantly improved wheat root attributes as compared to chemical fertilizers alone. This might be due to chemical fertilizers being in a quickly available form and plants easily adsorb them in sandy soil, while TW slowly decomposed and mineralized the nutrients for use throughout the plant and root growth stages. The highest TW rate (10%) showed a declined root growth as compared to 2.5% and 5%, which may be due to the higher microbial population and slow decomposition rate. The addition of chemical fertilizers could have been boosted microbial activities and accelerated the litter decomposition [34]. In general, TW addition displayed positive effects on RLR and RF, but RMR was decreased as compared to the control (Table 2). The organic mulching is favorable for seedling emergence and root proliferation [35] and hence can increase the crop growth.

### 3.3. Effect of Amendments on N, P, and K Uptake in Wheat Shoots and Roots

The results of our study confirmed that the soil amendment was considerably influenced by total N, P, and K contents in the wheat shoots and roots under the sandy soil environment. The greatest uptake of N, P, and K in shoots and roots was measured under T4 over the control treatment. The uptake of nutrients under soil stress conditions was observed under poultry litter treatment [36]. However, the present work showed that the highest uptake of N, P and K in shoots and roots were reported from T4 treatment. The microbial population increased after the incorporation of organic amendments and nutrients which were slowly available to plants but also depended upon microbial activity and carbon decomposition rate [37].

### 3.4. Soil Enzymes Activities

The dissimilarity of specific soil enzyme activities has been recognized as well as the differences in the organic amendments and their chemical properties, adsorption and complexation with soil properties [38,39]. The activity of β-glucosidase is related to the carbon cycle and is an important source of energy for micro-organisms [40]. Urease activities played an important role in the nitrogen cycle [38,39], and [40] established that the oxidative activity of soil microorganisms indicates dehydrogenase activity. Our results indicated that urease, β-glucosidase and dehydrogenase activities were increased with the combined application of TW + F as compared to the control. However, the chemical fertilizer application alone had no significant affect, as there was no difference noted between the fertilizer application and control treatments, except for urease activities (Figure 4). The enzyme activities were increased with an increasing rate of TW application. Jeong et al. [41] examined the idea that urease activities are related to soil pH with an ideal range of 7–8.2, which supports our results because soil pH decreased with an increasing TW amendment rate, and increased urease activities. A similar trend was detected by [42]. In the present study, there was a similar trend between urease, β-glucosidase, and dehydrogenase with soil pH and soil enzymes having a negative correlation (*p* < 0.01) with soil pH. The activities of β-glucosidase and dehydrogenase are an indication of the biological oxidation of organic compounds released by TW amendment. The biological oxidation of organic amendments can be controlled by the dehydrogenase activities and β-glucosidase activities involved in the hydrolysis of the organic applications. Moreover, β-glucosidase activities release low molecular weight sugar, a major source of energy for soil microorganisms [39].

### 3.5. Effect of Amendment on Soil Chemical Properties

In our study, soil chemical properties were improved with the amendment rate of TW combined with chemical fertilizer as compared to fertilizer alone and the control. Soil pH displayed a decreasing trend with increased TW application rate, and chemical fertilizer treatment alone was slightly decreased over the control. Our results agreed with [39,42] that in soil treated with conocarpus waste and organic manure, soil pH was decreased as compared to control treatment. In our study, TW released acidic organic decomposable compounds, which could be the reason for the decreasing trend of soil pH. The application of organic amendments decreased soil pH; it might be that soil microbial activities help to decompose organic matter and release organic acids. This depended upon the soil and organic amendments properties as well as the decomposition rate [43].

The influences of TW amendment on the soil fertility status after the wheat harvested are as follows: the soil organic carbon (SOC) concentration was significantly increased with an increasing TW level. Similar findings were presented by [4] that the maximum SOC increased with increasing rates of different types of plant waste material and organic application under sandy soil [43]. In the present study, the highest SOC was noted for the 10% incorporation of TW. The different organic amendment combined with chemical fertilizer applications increased the SOC contents after the wheat was harvested under a pot experiment within the alkaline soil. SOC is major factor for nutrients and their availability for plant growth [8,44].

A large number of studies had investigated that plant growth, root growth traits and macronutrient availability, especially for the low mobility of nutrients in soil (P and K) improved with the application of organic amendments combined with chemical fertilizers [4,8]. In line with our expectations, we found an increasing trend in mineral nitrogen (NO_3_^−^-N and NH_4_^+^-N), available phosphorus (AP) and available potassium (AK) under co-application of TW and chemical fertilizer against the control treatment. The greater availability of soil nutrients was determined for T5 as compared to T1. Our results are consistent with the findings of [45] who evaluated that a higher availability of mineral nitrogen and available P was noted in the integrated application of sheep manure and chemical fertilizers as compared to the application of chemical fertilizer alone. The combined application of organic amendments and chemical fertilizers significantly increased the soil nutrients such as N, P, and K over the chemical fertilizers alone [44].

## 4. Materials and Methods

### 4.1. Collection of Milk Tea Waste (TW) and Sandy Soil

Milk tea waste was collected from a milk tea restaurant at Xi’an, China and shipped to the Soil Physics laboratory of Northwest A&F University. Milk tea waste was air dried for one week, washed with distilled water and oven dried at 70 °C for 48 h and ground to pass a 2-mm sieve. A sandy desert soil (0–20 cm) was collected from the hinterland of Taklimakan Desert (39°00′ N and 83°40′ E, with an altitude of 1099 m). The collected soil sample was stored in polyethylene bags and transferred to the laboratory and unwanted material was removed (roots and litters). The sample was air dried and sieved through a 2-mm sieve to be used for further analysis. The basic characteristics of the soil and TW used in this study are mentioned in Table 4.

### 4.2. Experimental Setup

The pot experiment was designed as a randomized complete block, containing five treatments with three replicates for the sampled sandy soil. Three levels of milk tea waste (TW) combined with chemical fertilizers (F) were used in the experiment. Five treatments included the control without an amendment (T1); chemical fertilizers (F) (T2, 0.06 g·kg^−1^ N, 0.03 g·kg^−1^ P, and 0.06 g·kg^−1^ K); TW 2.5% (T3, 25 g·kg^−1^ + F); TW 5% (T4, 50 g·kg^−1^ + F); and TW 10% (T5, 100 g·kg^−1^ + F). The chemical fertilizers N, P, and K were applied as a source of urea, di-ammonium phosphate (DAP) and sulfate of potash (SOP). For each treatment, 500 g of sandy soil and the proper amendment was homogenized manually and transferred into each pot. The sandy soil in each plastic pot was moistened to about 70% water holding capacity (WHC) by adding distilled water. After amendment incorporation, five wheat seeds were sown in each pot. The pots were properly arranged in a completely randomized design at the greenhouse and water loss was checked every day, to maintain a 70% WHC on a daily basis up to end of the experiment. The indoor greenhouse temperature was maintained at 25 ± 2 °C during the experimental time period. After 10 days, wheat plants were thinned and left three healthy plants in each pot, and after eight weeks the wheat plants were uprooted from each pot, and the parameters of aboveground biomass and roots were determined.

### 4.3. Plant Growth Parameters

After eight weeks of sowing, wheat plants were uprooted from each pot, and then aboveground (stem and leaves) and belowground part (roots) were thoroughly washed with tap water and then by distilled water for the removal of adhered soil and milk tea waste particles properly. Both roots and aboveground biomass were properly dried with tissue paper and growth parameters such as chlorophyll content, leaf area index, plant height, plant fresh and dry weight, and root fresh and dry weight were measured. Leaf area was calculated by the standard formula given by [46].
Leaf area = L × W × A
where L is the leaf length, W is the leaf width and A is the indicating factor value (0.80) for the used wheat crop.

#### 4.3.1. Chlorophyll Content

The chlorophyll content was measured by using the standard protocol described by [47]. Fresh leaves of wheat (0.2 g) were placed in 25 mL of 80% acetone and left at room temperature for 48 h in the dark. The absorbance of the extract was measured using a spectrophotometer (UV-2450, Shimadzu, Kiyoto, Japan) at different wavelengths of 663 nm and 645 nm. The chlorophyll a, b, and carotenoid contents were calculated by the formula given by [47].

#### 4.3.2. Root Traits

Root length, root surface area, root volume and diameter were measured using a Hewlett Packard scanner controlled by Win-RHIZO, 2007d (Reagent Instruments Inc. Ltd., Model- J221 A, Seiko Epson corporation, Japan) software to measure the root parameters. Roots were separated and the surface area of roots and projected area of roots were washed with distilled water. Roots were placed in a plexiglass tray (200 × 300 mm) with a 4–8 mm deep water layer but water was added according to root size, using an EPSON PREFECTIONTM V700 Photo Flatbed Scanner (Delta-T Area Meter Type AMB2; Delta-T Device Ltd., Cambridge, UK)—6400 dpi × 9600 dpi.

### 4.4. Chemical Analysis of Soil, Plant, and Milk Tea Waste

Soil electrical conductivity (EC) and pH were measured in a 1:2.5 (*w*/*v*) soil-water extract, and 1:10 (milk tea waste: H_2_O) extract using a glass electrode (DDS-307 EC meter, Shanghai Bante Instrument Co., Ltd., China) and a pH meter (Mettler Toledo 320-S, Shanghai Bante Instrument Co., Ltd., Shanghai, China). A soil particle size analysis was determined using a Master sizer 2000E (Malvern, UK) laser diffract meter [48]. Soil organic carbon (SOC) was determined following wet digestion with sulfuric acid (H_2_SO_4_) and potassium dichromate (K_2_CrO_7_). Total nitrogen (TN), total organic carbon (TOC) and C:N of milk tea waste (TW) were determined by using a CN Analyzer (Vario Max, Elementar, Germany). Total N, P, and K contents of shoots and roots of wheat and TW were analyzed colorimetrically after digestion with (H_2_SO_4_ + HClO_4_), briefly described by [49]. Soil NH_4_^+^ and NO_3_^−^ for each fresh soil sub-sample was extracted with 2 M KCl (soil: solution 1:10), shaken for 1h to determine the concentration of mineral nitrogen using a continuous flow analyzer (Bran and Luebbe AA3, Norderstedt, Germany). Available P within the soil was measured using a 0.5 M NaHCO_3_ (pH 8.5) extract followed by a visible light spectroscopic analysis of a blue colored complex according to the method in (UV-VIS spectrophotometer, Model UV-2450, Shimadzu, Kiyoto, Japan) [50]. Available K was determined using 1 N ammonium acetate (NH_4_OAc) extraction followed by emission spectroscopy (FP 6410, Shanghai Bante Instrument Co., Ltd., China) [51]. Total zinc (Zn), copper (Cu), and iron (Fe) content of milk tea waste (TW) analyzed by digested using HNO_3_-HClO_4_ (3:1) and the resultant solution was analyzed for total concentration of Zn, Cu, and Fe contents by using USEPA Method 3051A by AAS, Z-2000, Japan [52]. All above mentioned analysis were repeated three times.

### 4.5. Soil Enzyme Assays

The enzyme activities of urease, dehydrogenase and β-glucosidase were analyzed by using the colorimetric method (UV-VIS spectrophotometer, Model UV-2450, Shimadzu, Kiyoto, Japan) [53]. For the determination of urease activity, 5 g of moist soil was incubated at 37 °C for 12 h after adding (NH_4_^+^) released from a solution of urea (10%), and the control was processed by adding distilled water. Dehydrogenase activity was determined based on extraction with methanol (CH_3_OH) and the colorimetric determination of the triphenylformazan (TPF) made from the reduction of 2,3,5-triphenyltetrazolium chloride (TTC). β-glucosidase activity was determined based on the colorimetric determination of *p*-nitrophenyl released by β-glucosidase, and 1 g of air-dried soil was incubated for 1 h at 37 °C with a buffered PNG (*p*-nitrophenyl-β-d glucosidase) solution (pH 6.0) and toluene.

### 4.6. Statistical Analysis

Significant differences in plant growth, root traits and soil properties were statistically analyzed by using SPSS 22 software (SPSS Inc., Chicago, IL, USA), and figures were generated by using Origin Pro. 9.0 (Northampton, MA, USA). Experimental results of the pot experiment were presented as triplicate means ± SE by one–way analysis of variance (ANOVA). Mean comparison testing was performed using the least significant differences (*p* < 0.05). Patterns in the data were investigated using bivariate Pearson correlation tests. Principal component analysis (PCA) was used to evaluate the overall differences in the chemical fertilizer alone and chemical fertilizer with tea waste material, and also to test the relationship between root attributes and the N, P, and K contents of roots and shoots and to test the relationship between soil enzymes and soil properties.

## 5. Conclusions

This study emphasizes the practical usage of milk tea waste (TW) combined with chemical fertilizer as a soil nutrient source for better wheat growth, root traits, and soil biochemical properties within sandy soil. The results indicated that milk tea waste combined with chemical fertilizer treatment improved the growth performance of wheat plants (plant height, chlorophyll a, chlorophyll b, carotenoid, and shoot and root dry weight) and N, P and K contents in shoots and roots in comparison to the application of chemical fertilizer alone and control treatments. Among all treatments, T4 (TW 5% and F) most significantly improved wheat growth. All investigated soil biochemical properties except for soil pH were increased with the increasing amendment rates. Therefore, we concluded that the combined application of TW 5% and chemical fertilizer should be a viable nutrient source for land application, resulting in improved plant aboveground and belowground biomass as well as improved soil fertility status within sandy soil. Following this pot experiment, related field tests are needed and could be carried out in the future.

## Figures and Tables

**Figure 1 molecules-24-00423-f001:**
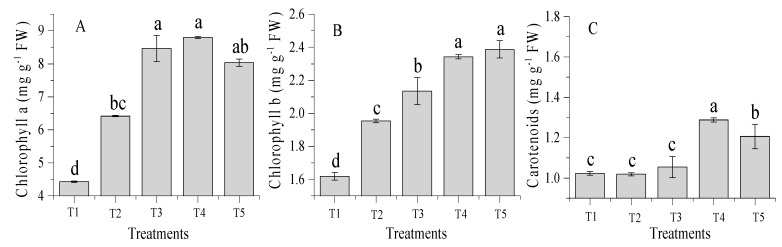
Effect of co-application of TW and chemical fertilizers on chlorophyll a (**A**) and b (**B**) and carotenoid (mg·g^−1^) (**C**) content in wheat leaves under without amendment (T1), chemical fertilizers (T2), TW 2.5% + F (T3), TW 5% + F (T4) and TW 10% + F (T5). The data showed the mean of three replications, and error bars are standard deviations. Different letters indicate there were significant differences (*p* < 0.05) in the LSD means comparisons between the treatments mean.

**Figure 2 molecules-24-00423-f002:**
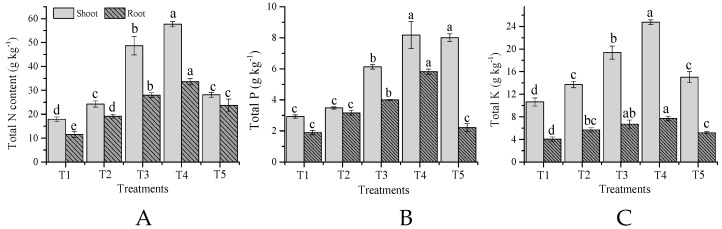
Influences of co-application of TW and chemical fertilizers on N (**A**), P (**B**), and K (**C**) uptake in shoot and root of wheat. Without amendment (T1), chemical fertilizers (T2), TW 2.5% + F (T3), TW 5% + F (T4) and TW 10% + F (T5). Data showed the mean of three replications, and error bars are standard deviations. Different letters indicate there were significant differences (*p* < 0.05) in the LSD means comparisons between the treatments mean.

**Figure 3 molecules-24-00423-f003:**
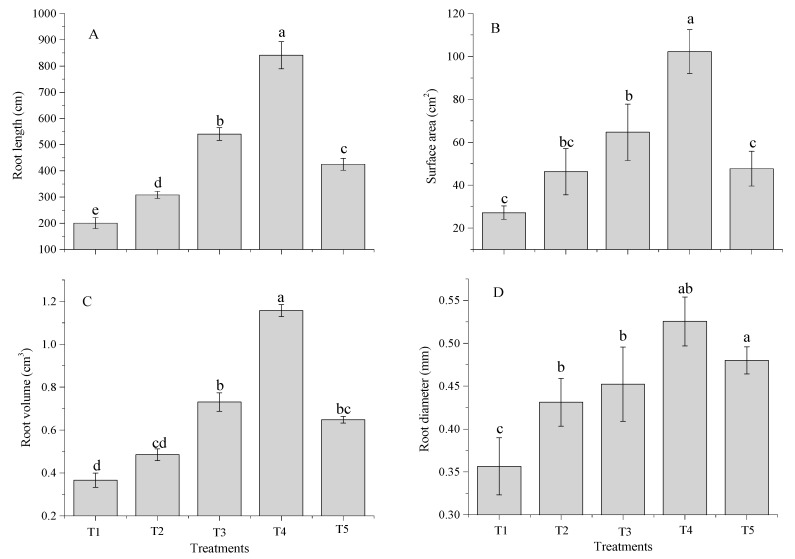
Influences of co-application of TW and chemical fertilizers (root length (**A**), root volume (**C**), surface area (**B**), and root diameter (**D**). Without amendment (T1), chemical fertilizers (T2), TW 2.5% + F (T3), TW 5% + F (T4) and TW 10% + F (T5). The data showed the mean of three replications, and error bars are standard deviations. Different letters indicate there were significant differences (*p* < 0.05) in the LSD means comparisons between the treatments mean.

**Figure 4 molecules-24-00423-f004:**
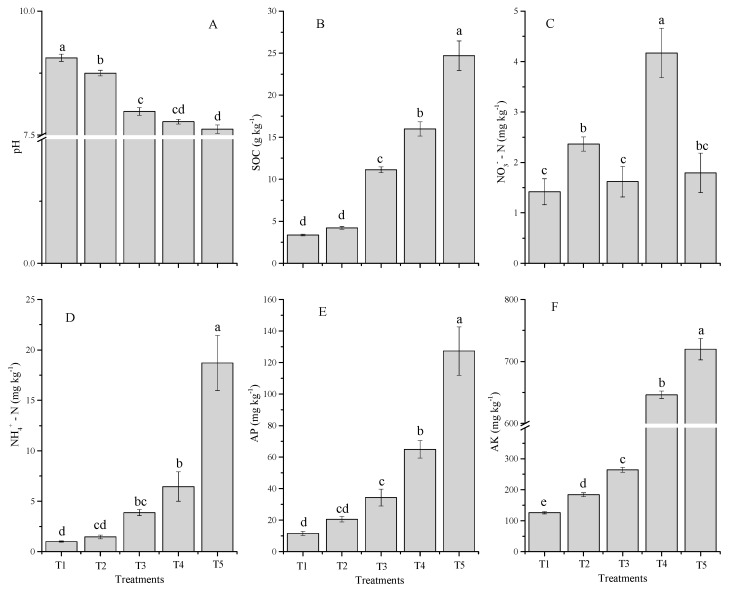
Influences of co-application of TW and chemical fertilizers soil chemical properties, soil pH (**A**), SOC (**B**), NO_3_^−^-N (**C**), NH_4_^+^-N (**D**), AP (**E**) and AK (**F**) after wheat harvested. Without amendment (T1), chemical fertilizers (T2), TW 2.5% + F (T3), TW 5% + F (T4) and TW 10% + F (T5). The data showed the mean of three replications, and error bars are standard deviations. Different letters indicate there were significant differences (*p* < 0.05) in the LSD means comparisons between the treatments mean.

**Figure 5 molecules-24-00423-f005:**
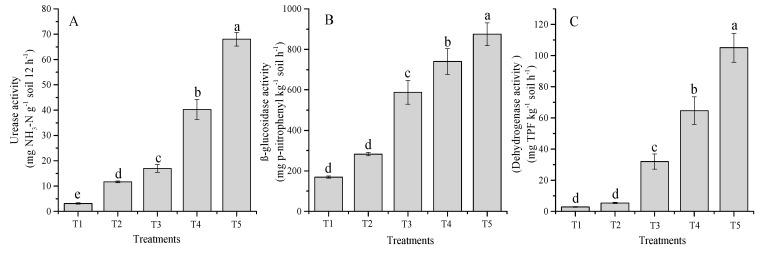
Influences of co-application of TW and chemical fertilizers on soil enzymes activities, urease (**A**), β-glucosidase (**B**) and dehydrogenase (**C**) after wheat harvested. Without amendment (T1), chemical fertilizers (T2), TW 2.5% + F (T3), TW 5% + F (T4) and TW 10% + F (T5). The data showed the mean of three replications, and error bars are standard deviations. Different letters indicate there were significant differences (*p* < 0.05) in the LSD means comparisons between the treatments mean.

**Figure 6 molecules-24-00423-f006:**
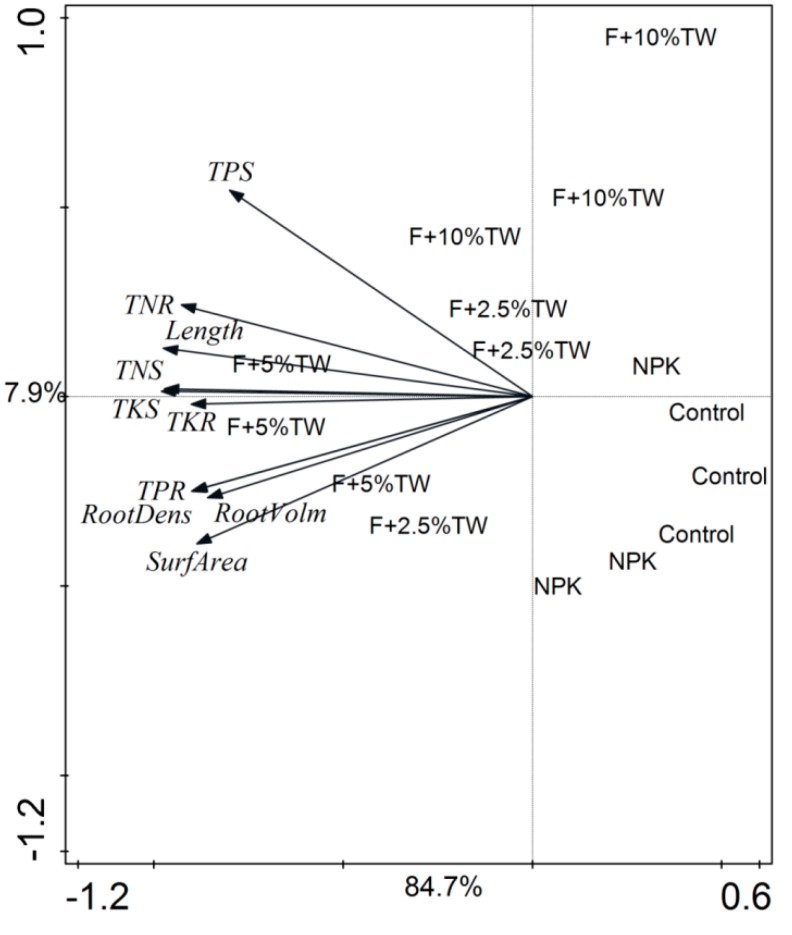
Principal component analysis (PCA) of root traits and plant NPK content of wheat crop. TNS (total nitrogen shoot), TNP (total nitrogen root), TPS (total phosphorus shoot), TPR (total phosphorus root), TKS (total potassium shoot), and TKR (total potassium root).

**Figure 7 molecules-24-00423-f007:**
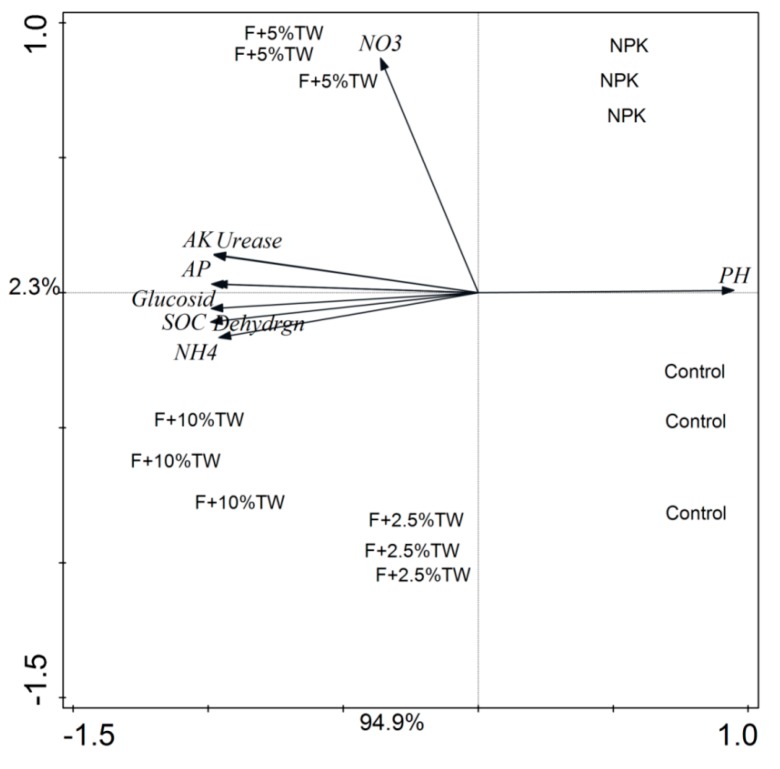
Principal component analysis (PCA) of soil enzymes and soil chemical properties after wheat harvested.

**Figure 8 molecules-24-00423-f008:**
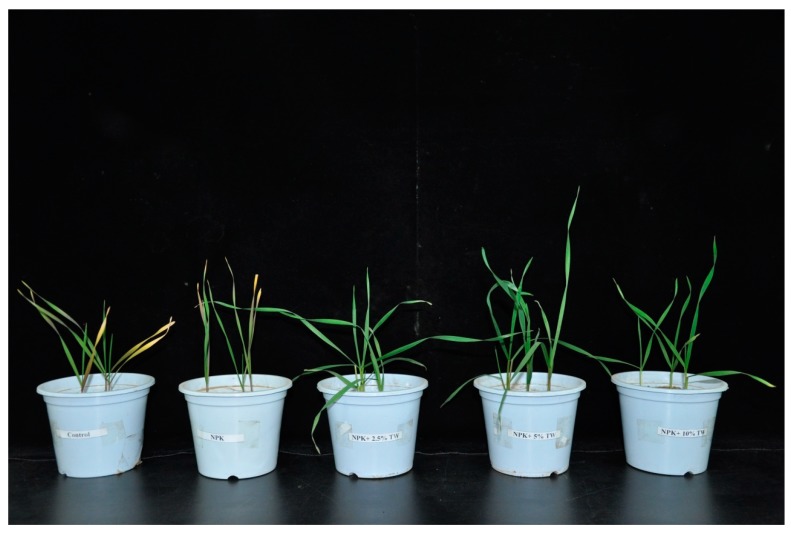
Influences of co-application of TW and chemical fertilizers on physiological parameters of wheat plants after 56 days. T1 (Control without an amendment), T2 (Chemical fertilizers), T3 (F + 2.5% TW), T4 (F + 5% TW), and T5 (F + 10% TW).

**Table 1 molecules-24-00423-t001:** Influence of chemical fertilizers alone and combined with milk tea waste (TW) on plant height, shoot, and root dry weight and leaf ar ea index parameters after 8 weeks (mean ± standard error; *n* = 3).

Treatment	Plant Height (cm)	Shoot Dry Weight (g·pot^−1^)	Root Dry Weight (g·pot^−1^)	Leaf Area Index (cm^2^)
T1	18.0 ± 1.6 ^d^	0.14 ± 0.01 ^c^	0.16 ± 0.01 ^b^	0.30 ± 0.01 ^c^
T2	20.5 ± 0.3 ^c^	0.23 ± 0.02 ^c^	0.19 ± 0.01 ^b^	0.41 ± 0.05 ^bc^
T3	25.0 ± 0.6 ^b^	0.33 ± 0.01 ^b^	0.28 ± 0.03 ^a^	0.62 ± 0.06 ^b^
T4	29.5 ± 0.6 ^a^	0.50 ± 0.07 ^a^	0.33± 0.03 ^a^	0.87 ± 0.06 ^a^
T5	19.2 ± 1.0 ^c^	0.25 ± 0.03 ^bc^	0.21 ± 0.02 ^b^	0.57 ± 0.01 ^b^

Without amendment (T1), chemical fertilizers (T2), TW 2.5% + F (T3), TW 5% + F (T4) and TW 10% + F (T5). Different letters indicate there were significant differences (*p* < 0.05) in the LSD means comparisons between treatments mean.

**Table 2 molecules-24-00423-t002:** Influence of chemical fertilizers and combined with milk tea waste (TW) on root traits after eight weeks (mean ± standard error; *n* = 3).

Treatment	Root Length Density (cm·cm^−3^)	Root Length Ratio (cm·g^−1^)	Root Mass Ratio (g·g^−1^)	Root Fineness (cm·cm^−3^)
T1	0.43 ± 0.06 ^b^	674 ± 41 ^c^	0.53 ± 0.01 ^a^	457 ± 62 ^c^
T2	0.67 ± 0.05 ^b^	683 ± 63 ^c^	0.42 ± 0.04 ^bc^	525 ± 49 ^c^
T3	1.23 ± 0.22 ^a^	928 ± 30 ^ab^	0.45 ± 0.05 ^abc^	848 ± 30 ^ab^
T4	1.31 ± 0.30 ^a^	1145 ± 137 ^a^	0.41 ± 0.01 ^c^	956 ± 80 ^a^
T5	0.75 ± 0.08 ^b^	801 ± 16 ^bc^	0.51 ± 0.03 ^ab^	789 ± 16 ^b^

Without amendment (T1), chemical fertilizers (T2), TW 2.5% + F (T3), TW 5% + F (T4) and TW 10% + F (T5). Different letters indicate there were significant differences (*p* < 0.05) in the LSD means comparisons between the treatments mean.

**Table 3 molecules-24-00423-t003:** Bivariate correlation test between physicochemical properties and soil enzymes activity after wheat plants harvested.

	pH	SOC	NO_3_^−^	NH_4_^+^	AP	AK	Urease	Glucosidase	Dehydrogenase
pH	1	−0.839 **	−0.399	−0.641 **	−0.730 **	−0.817 **	−0.760 **	−0.934 **	−0.812 **
SOC		1	0.209	0.928 **	0.950 **	0.942 **	0.975 **	0.944 **	0.985 **
NO_3_^−^			1	0.046	0.129	0.485	0.267	0.309	0.248
NH_4_^+^				1	0.926 **	0.837 **	0.950 **	0.780 **	0.913 **
AP					1	0.905 **	0.969 **	0.877 **	0.950 **
AK						1	0.955 **	0.915 **	0.954 **
Urease							1	0.896 **	0.975 **
Glucosidase								1	0.921 **
Dehydrogenase									1

** Correlation is significant at the 0.01 level (two-tailed).

**Table 4 molecules-24-00423-t004:** Properties of soil and milk tea waste used in the experiment (mean ± standard error; *n* = 3).

Parameters	Soil	Milk Tea Waste (TW)
Clay (%)	5.50	
Silt (%)	15.61	
Sand (%)	78.89	
Water holding capacity (WHC)	16%	
pH (1:2.5)	9.21 ± 0.5	5.36 ± 0.3 (1:10)
EC (1:2.5) (μS·cm^−1^)	1513 ± 40.7	1600 ± 48.5 (1:10)
Total carbon (%)	1.97 ± 0.05	48.10 ± 4.5
Total N (%)	0.02 ± 0.0	3.76 ± 0.5
C:N	84.92 ± 4	12.79 ± 2
Total P (%)		0.42 ± 0.01
Total K (%)		0.52 ± 0.01
Olsen P (mg·kg^−1^)	12.15 ± 1	
K exchangeable (mg·kg^−1^)	125.56 ± 3	
Total Zn (mg·kg^−1^)		34.20 ± 2.1
Total Cu (mg·kg^−1^)		6.05 ± 0.9
Total Fe (mg·kg^−1^)		774.34 ± 18.5

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
