# Peer review of "Co-Application of Milk Tea Waste and NPK Fertilizers to Improve Sandy Soil Biochemical Properties and Wheat Growth"

_molecules, 2019, doi:10.3390/molecules24030423_

Round 1

Reviewer 1 Report

The manuscript contains interesting data set and good results...

Please, consider some suggestions to improve the manuscript edition:

1) Please revise all English writing and grammar along the text...

2) Consider as a sugtestion the following title: Co-application of milk waste and NPK fertilizers to improve sandy soil biochemical properties and root and wheat growth

2) Introduction should revised to include the effects of treatments on root, wheat growth and soil biological properties already publiched....

3) Objectives of the study are not clearly identified... Reconsider, please, the edition....

4) Please, define the criteria used to choose milk tea waste range rates....

5) Levels of N P e K ar tow to grow plants in small pots and for long time... Pleasecite the literature which support the NPK recommendations...

6) Discussion of results should be improved to enclose all references already for milk and other wastes used in agricultures and the role played in the issues investigated.

7) Please use, dose or rate as words to replace dosage...

8) In the item conclusions, please edit the text and answer the following questions: which are the best treatment for roots, wheat growth and sandy soil biochemical properties improvements? Is the best treatment viable to be use in field and to attend local legislation regarding the use of animal wastes in crop fields...

Thank you for the oportunity to revise this manuscript.

Author Response

REVIEWER: 1

Comments and Suggestions for Authors

The manuscript contains interesting data set and good results.

Please, consider some suggestions to improve the manuscript edition: 

Comment 1. Please revise all English writing and grammar along the text.

Response: As per suggestion we have improved English writing by the Editing Company, and all the changes have been with track changes and highlighted with (Color).

Comment 2. Consider as a suggestion the following title: Co-application of milk waste and NPK fertilizers to improve sandy soil biochemical properties and root and wheat growth.

Response: Thank you for suggestion; we have revised the title.

Comment 3. Introduction should revised to include the effects of treatments on root, wheat growth and soil biological properties already published.

Response: We have improved the whole introduction section.

Comment 4. Objectives of the study are not clearly identified... Reconsider, please, the edition.

Response:  Thank you for valuable suggestion; we have thoroughly revised the objectives (Page No. 3, Line No. 105-109).

Comment 5. Please, define the criteria used to choose milk tea waste range rates.

Response: In the present study we used milk tea waste, which contains nutrients such as N, P, K and a low C:N ratio as compared to other organic amendments (Page No. 3 and Line No. 103-104). However, soil was low contents of N, P, K and C, that is why we applied different rate with chemical fertilizer. 

Comment 6.  Levels of N P e K ar tow to grow plants in small pots and for long time... Please, cite the literature which support the NPK recommendations.

Response: According to milk tea waste nutrient contents we have applied the low fertilizer rates.

Comment 7. Discussion of results should be improved to enclose all references already for milk and other wastes used in agricultures and the role played in the issues investigated.

Response: As advised, we improved the whole discussion section and added new reference. (Line No. 383-384).

Comment 8. Please use, dose or rate as words to replace dosage

Response: As per suggestion we have replaced the dosage with rate throughout the paper.

Comment 9. In the item conclusions, please edit the text and answer the following questions: which are the best treatment for roots, wheat growth and sandy soil biochemical properties improvements? Is the best treatment viable to be use in field and to attend local legislation regarding the use of animal wastes in crop fields.

Response: Thank you very much for valuable suggestion, we have improved conclusion and written the best treatment in conclusion section (Line No.542-553).

Reviewer 2 Report

In this work, the authors investigate the effects of  co-application of milk tea waste and inorganic fertilizer on sandy soil biochemical properties and wheat growth. The purpose of the paper is clearly stated and  the manuscript is well organized. The  statistical analysis and data are appropriated. Minor comments are reported in the pdf file and:    

1) I suggest to modify the title of the manuscript in " Co-application of milk tea waste and inorganic fertilizer improve sandy soil biochemical properties and wheat growth"

2) The bulk of Introduction section is ok but there are some concepts not clear that can be improved.

3) I suggest to the author to move the Figure 4 in the first paragraph of the Result section "2.1. Effect of milk tea waste application on plant growth" adding a short description.

4) The bulk of Results section is ok. I suggest to make some modification as reported in the pdf file.

5) Could you insert in the M&M the greenhouse conditions?

6) Did you conduct some other analysis on the milk tea waste? I believe that the waste that you used in the experiment is not always the same, so it could be useful to have some other information on it. Finally, the authors conducted only one experiment, so it could be interesting to observe the repeatability of the results.

Author Response

 REVIEWER: 2 In this work, the authors investigate the effects of co-application of milk tea waste and inorganic fertilizer on sandy soil

biochemical properties and wheat growth. The purpose of the

paper is clearly stated and the manuscript is well organized.

The statistical analysis and data are appropriated. Minor comments

are reported in the pdf file and: Comment 1: I suggest to modify the

title of the manuscript in "Co-application of milk tea waste and inorganic fertilizer improve sandy

soil biochemical properties and wheat growth"

Response: Thank you for fruitful advised, we have modified the

title of the manuscript (Page no. 1, Line No. 5-7). 

Comment 2: The bulk of Introduction section is ok but there

are some concepts not clear that can be improved.

Response: As fruitful suggestion, we have thoroughly

revised the introduction with new additions at several

places in order to address the aforementioned points. We

also have made several changes in the introduction section in

order to make the significance of our study more clear and focused.

Comment 3: I suggest to the author to move the Figure 4 in the

first paragraph of the Result section "2.1. Effect of milk tea

waste application on plant growth" adding a short description.

Response: According to advised, we have moved the Figure 4 in

the first paragraph of the Result section "2.1.Effect of milk tea

waste application on plant growth" added a short description.

 Comment 4: The bulk of Results section is ok. I suggest to make

some modification as reported in the pdf file.

Response:  As per suggestion we have modified according

to pdf file comments and improved throughout the paper.

Comment 5: Could you insert in the M&M the greenhouse conditions.

Response:  As per comment, we have inserted the greenhouse

conditions. The indoor greenhouse temperature was maintained

at 25±2 0C during the experimental time period (476-477).

Comment 6: Did you conduct some other analysis on the milk tea

waste? I believe that the waste that you used in the experiment is

not always the same, so it could be useful to have some other

information on it.

Response:  Thanks for valuable suggestion, we have analyzed the

milk tea waste some other analysis which have mentioned in Table 1.

Comment 7: Finally, the authors conducted only one experiment, so

it could be interesting to observe the repeatability of the results.

Response: Thanks a lot for suggestions, we were conducted the

experiment in control conditions and all analysis measured the standard

protocol.

Round 2

Reviewer 1 Report

Dear Editor, after reading the new version of the manuscript, in my opinion, the document can be accepted for publication after minor English and editing review. Please, consider the possibility to describe in the conclusion which is the best treatment or if there is a specific optimum combination of milk tea and fertilizer for each group of attributes investigated. 

Best regards.

Author Response

REVIEWER: 1

Comments and Suggestions for Authors

After reading the new version of the manuscript, in my opinion, the document can be accepted for publication after minor English and editing review. Please, consider the possibility to describe in the conclusion which is the best treatment or if there is a specific optimum combination of milk tea and fertilizer for each group of attributes investigated.

Response: Thanks a lot for suggestions, we have described in the conclusion, the best treatment which is a specific optimum combination of milk tea waste and chemical fertilizers for wheat growth and root traits and as well as biochemical properties of sandy soils (Line No. 508-511). We improved the whole paper and corrected the English error.